# Rational Design and Synthesis of HSF1-PROTACs for Anticancer Drug Development

**DOI:** 10.3390/molecules27051655

**Published:** 2022-03-02

**Authors:** Chiranjeev Sharma, Myeong A Choi, Yoojin Song, Young Ho Seo

**Affiliations:** College of Pharmacy, Keimyung University, Daegu 704-701, Korea; 1chiranjeevsharma1@gmail.com (C.S.); auddk5495@naver.com (M.A.C.); suj741741@gmail.com (Y.S.)

**Keywords:** HSF1, KRIBB11, PROTAC, thalidomide, anticancer drug development

## Abstract

PROTACs employ the proteosome-mediated proteolysis via E3 ligase and recruit the natural protein degradation machinery to selectively degrade the cancerous proteins. Herein, we have designed and synthesized heterobifunctional small molecules that consist of different linkers tethering KRIBB11, a HSF1 inhibitor, with pomalidomide, a commonly used E3 ligase ligand for anticancer drug development.

## 1. Introduction

Cancer continues to be one of the leading causes of morbidity and mortality worldwide, accounting for 10 million deaths in 2020 [1]. It has an immense economic impact, which is escalating, in addition to the huge loss of human lives. According to the WHO report, the total annual economic cost of cancer in 2010 was estimated at approximately USD 1.16 trillion [2]. The existing treatment by traditional chemotherapy, targeted therapy, and immunotherapy is fraught with the disadvantages of severe adverse reactions and toxicity, resistance to the treatment therapy, and excessive uncontrollable activity of the immune system [3,4,5,6]. Thus, new strategies to combat cancer are an urgent medical need.

Targeting proteins that contribute to cancer development for therapeutic purposes is challenging, and most of the proteome is still considered beyond reach that is deemed undruggable. Sakamoto et al. reported a novel chemical strategy in 2001, which can be used to rationally manipulate the intracellular levels of many proteins that are regulated by the molecular mechanism of ubiquitin-dependent proteolysis [7]. This technology opened new avenues in anticancer drug discovery and was also commercialized as PROTAC^®^ for targeted protein degradation [8]. Proteolysis-targeting chimeras are heterobifunctional molecules that can target any protein for ubiquitination and subsequent proteasomal degradation [9]. Further, lysosomal targeting with lysosomal-targeting chimeras (LYTACs), autophagy targeting with autophagy-targeting chimeras (AUTACs), and RNA targeting with ribonuclease-targeting chimeras (RIBOTACs) have been reported by recruiting alternative degradation pathways [10].

HSF1 is a master regulator of transcriptional responses to proteotoxic stress and an effective drug target for anticancer therapies including radiotherapy and thermotherapy [11]. HSF1 is known to be overactive or overexpressed in many cancer types including prostate cancer, pancreatic cancers, breast cancer, colorectal, lymphomas, melanoma, oral cancers, and so forth [12]. The activation of HSF1 essentially results in the over-expression of various heat shock proteins, which affects the outcome of the treatment.

The over-expression of heat shock proteins caused by HSF1 is reported as the main cause of resistance to chemotherapy, radiation therapy, and hypothermia used in cancer treatment [13]. Although HSF1 has been found to be a significant factor in the development and low prognosis of various chemotherapy cancer treatments, there are still very few effective inhibitors of HSF1 [14]. Most HSF1 inhibitors reported to date are indirect inhibitors through high signal disturbance in HSF1 rather than directly inhibiting HSF1. Continuing our effort of developing anticancer drug candidates [15,16], we report the rational design and synthesis of HSF1-PROTACs in an endeavor to create new chemical agents for the development of appropriate clinical candidates in anticancer drug discovery.

## 2. Results and Discussion

Our design rationale for making PROTAC involved tagging a known binder of HSF1 protein. A 3-nitropyridine derivative (N2-(1H-indazole-5-yl)-N6-methyl-3-nitropyridine-2,6-diamine) named KRIBB11 is a well-known HSF1 inhibitor used to treat tumor tissues without significant body weight loss [17]. The docking between the crystal structure HSF1 [18] and KRIBB11 revealed that the hydrogen bonding by the oxygen atoms of the nitro group of the inhibitor stabilizes the binding to HSF1 protein. The nitrogen atoms of the pyridine ring and the N-methyl group of the inhibitor form hydrogen bonding interactions with the amide group of Gln72 and the phenol group of Tyr76, respectively. The N-methyl moiety in KRIBB11 is projected away from the protein, as shown in Figure 1. Thus, we decided to link the E-3 ligase ligand to the N-methyl part of KRIBB11. We employed three different linkers. Phthalimide conjugation is the most effective strategy for proteasome-mediated proteolysis [19]. Accordingly, we designed the PROTAC molecules by connecting KRIBB11 and pomalidomide as a ligand for E3 ligase.

We began the synthesis for making the KRIBB11 part by treating 2,6-dichloro-3-nitropyridine with 5-aminoindazole in the presence of triethylamine in ethanol, resulting in the formation of **3** in good yield [20]. Subsequent reaction with 1,4-diaminobutane in acetonitrile at room temperature yielded **4**, as shown in Figure 1. The pomalidomide part **9c** and a precursor **9a** was synthesized using the previously reported procedure, as illustrated in Figure 2 [21].

The linkers were connected to the pomalidomide, as shown in Figure 3. Firstly, the 4,7,10-Trioxatridecane-1,13-diamine **10a** was mono-Boc protected into **10b** using Boc anhydride and triethylamine in anhydrous dichloromethane. It was then treated with **9a** in the presence of diisopropylethylamine in DMF to form **12a**, which was further reacted with TFA to give **12b** in good yield. The mono Boc-protected 4,7,10-Trioxatridecane-1,13-diamine **10b** was also reacted with chloroacetyl chloride in the presence of Cs_2_CO_3_ to give **11**. It was then treated with **9c** to form **13a**, which, on subsequent TFA treatment, gave **13b**. Finally, the HSF1-PROTACs were synthesized by nucleophilic aromatic substitution, as shown in Figure 4. The PROTAC **14** was synthesized by heating **4** and **9a** in DMF in the presence of diisopropylethylamine. The PROTACs **15** and **16** were synthesized by reacting **3** at room temperature with **12b** and **13b**, respectively, in the presence of triethylamine in acetonitrile. The copies of ^1^H and ^13^C spectra for all prepared compounds are given in Appendix A. PROTACs were purified by MPLC, and the purity was also determined by HPLC (Appendix A).

With compound **14**–**16** in hand, we examined the dose-dependent effect of **14**–**16** on the cell viability of MDA-MB-231, which is a highly aggressive and invasive triple negative breast cancer (TNBC) cell line. The preliminary investigation suggested **14**–**16** exhibited a minimal in vitro inhibitory effect against MDA-MB-231 breast cancer cells, as shown by the cell viability data in Figure 2. None of compounds **14**–**16** could inhibit the 50% cell growth up to 50 μM treatment. Compound **15** provided a slightly better inhibitory effect than **14** and **16,** in that compound **15** impaired 27% cell growth by the treatment of 30 and 50 μM concentration.

We further evaluated the proteolysis effect of compounds 14–16 on HSF1 (Figure 3). Our Western blotting data indicated that **14** showed a little degradation of HSF1. However, the results were not consistent despite several repetitions. We speculated that one reason for inconsistency could be due to the intrinsic formation of Hsp90-Hsp70-HSF1 multi-chaperone complex in the cancer cell, hampering compound **4** to interact with the binding site of HSF1. HSF1 is dynamically associated or dissociated from multi-chaperone complex under various stressed or non-stressed conditions. Hence, we decided to treat cancer cells with compounds **14**–**16**, together with an Hsp90 inhibitor, which could induce a heat shock response (HSR) to dissociate HSF1 from the chaperone complex. Unfortunately, the results did not still show any meaningful degradation of HSF1 in MDA-MB-231 cells, shown in Figure 3B. To draw biological meaningful conclusions, our efforts are directed toward finding better PROTACs with new HSF1 ligands and E3 ligase ligands. The result will be reported in due course.

## 3. Materials and Methods

### 3.1. General Information

All reagents and solvents were purchased from commercial suppliers and used without further purification. All experiments dealing with moisture-sensitive compounds were carried out under argon atmosphere. Concentration or solvent removal under reduced pressure was carried out using rotary evaporator. Analytical thin layer chromatography was performed on precoated silica gel F254 TLC plates (E, Merck) with visualization under UV light or by staining using iodine. Column chromatography and medium-pressure liquid chromatography (MPLC) was conducted on silica (Merck Silica Gel 40–63 m) or performed by using a Biotage SP1 flash purification system with prepacked silica gel cartridges (Biotage, Uppsala, Sweden). NMR analyses were carried out using a JNM-ECZ500R (500 MHz) manufactured by Jeol resonance. Chemical shifts are reported in parts per million (δ). The deuterium lock signal of the sample solvent was used as a reference, and coupling constants (J) are given in hertz (Hz). The splitting pattern abbreviations are as follows: s, singlet; d, doublet; t, triplet; q, quartet; dd, doublet of doublet; td, triplet of doublet; and m, multiplet. The purities of all final compounds were confirmed to be higher than 95% by analytical HPLC performed with a dual-pump Shimadzu LC-6AD system equipped with a VP-ODS C18 column (4.6 mm × 250 mm, 5 μm, Shimadzu, Kyoto, Japan). The LC-QTOF-MS analysis was performed using an Agilent 6530 Accurate-Mass Q-TOF LC/MS System with Agilent 1290 Infinity LC (Agilent Technologies, Palo Alto, CA, USA). The guard column and the analytical column were Zorbax SB-C8 (3.5 μm, 2.1 × 30 mm, Agilent Technologies) and Zorbax SB-Aq (1.8 μm, 2.1 × 100 mm, Agilent Technologies), respectively, and were maintained at 40 °C. The mobile phase consisted of 0.1% formic acid in water (A) and 0.1% formic acid in acetonitrile (B). The gradient conditions were as follows: 0–30 min, 1–20% B; 30–40 min, 20–90% B; 40–45 min, 90% B; 45–47 min, 90–1% B; and 47–52 min 1% B at a flow rate of 400 μL/min. The MS system was operated using ESI in the positive and the negative ionization mode. The optimized conditions of the QTOF-MS system for both ionization modes were as follows: drying gas temperature, 300 °C; drying gas flow, 10 L/min; nebulization pressure, 45 psi; sheath gas temperature, 350 °C; sheath gas flow, 10 L/min; capillary voltage, 3500 V; nozzle voltage, 0 V; fragmentor voltage, 175 V; and skimmer voltage, 65 V. The mass range was 50–1700 *m*/*z*, and the scan rate was 2.00 spectra/sec for both MS and MS/MS analyses.

### 3.2. Synthetic Procedures

#### 3.2.1. Synthesis of KRIBB11 Aanalogue 4

To 15 mL of ethanol were added 2,6-dichloronitropyridine (0.5 g, 2.59 mmol), 5-aminoindazole (0.36 g, 2.72 mmol), and triethylamine (0.4 mL, 2.85 mmol). The reaction mixture was refluxed for 90 °C for about 5 h. After the reaction was completed, the solid obtained was filtered, washed with methanol, and then dried under vacuum to afford compound 3 in 80.7% yield. ^1^H-NMR (500 MHz, DMSO-D_6_) δ 10.22 (s, 1H), 8.54 (d, J = 8.6 Hz, 1H), 8.09 (s, 1H), 7.92 (d, J = 1.7 Hz, 1H), 7.56 (d, J = 9.2 Hz, 1H), 7.44 (dd, J = 8.6, 1.7 Hz, 1H), and 6.95 (d, J = 8.6 Hz, 1H).

Compound **3** (0.289 g, 1 mmol) was dissolved in 25 mL acetonitrile, and 1,4-diaminobutane (0.44 mL, 5 mmol) was added. The reaction mixture was stirred at RT for 14 h. After completion of the reaction, acetonitrile was evaporated, and the residue was partitioned between 1N NaOH and ethyl acetate. The organic layer was dried with sodium sulfate and evaporated in vacuo to obtain compound **4** in 70.3% yield. ^1^H-NMR (500 MHz, DMSO-D_6_) δ 10.94 (s, 1H), 8.20 (s, 1H), 8.07 (d, J = 9.2 Hz, 1H), 8.04 (s, 1H), 7.53 (s, 2H), 6.09 (d, J = 9.7 Hz, 1H), 3.30 (t, J = 7.4 Hz, 2H), 2.48 (d, J = 6.9 Hz, 2H), 1.53 (t, J = 7.4 Hz, 2H), and 1.33 (t, J = 7.4 Hz, 2H).

#### 3.2.2. Synthesis of E-3 Ligase Lligands (**9a** and **9c**)

Synthesis of **9a**. A mixture of N-Boc-L-asparagine (2 g, 8.61 mmol), CDI (1.39 g, 8.61 mmol), and catalytic amounts 4-DMAP in anhydrous THF (21.5 mL) was stirred and heated to reflux at 76 °C for 48 h. After the end of the reaction, the mixture was filtered, and the solid was washed with THF to give 6 in 73% yield as a colorless solid. ^1^H-NMR (500 MHz, DMSO-D_6_) δ 10.77 (s, 1H), 7.15 (d, J = 8.6 Hz, 1H), 4.20–4.25 (m, 1H), 2.71 (dt, J = 17.8, 6.3 Hz, 1H), 2.46 (t, J = 3.7 Hz, 1H), 1.87–1.94 (m, 2H), and 1.40 (s, 9H).

Compound **6** (1.54 g, 6.75 mmol) was dissolved in TFA (7.8 mL, 101.25 mmol) and stirred for 30 min. The excess of the acid was removed in vacuo, and the resulting product was dried under vacuum to give **7** in quantitative yield as an off-white solid. ^1^H-NMR (500 MHz, DMSO-D_6_) δ 11.25 (s, 1H), 8.58 (s, 3H), 4.21 (d, J = 8.6 Hz, 1H), 2.67–2.75 (m, 1H), 2.59 (dt, J = 15.3, 2.3 Hz, 1H), 2.14–2.19 (m, 1H), and 2.02 (td, J = 13.0, 4.8 Hz, 1H).

Briefly, a mixture of 3-fluorophthalic anhydride (1.25 g, 7.5 mmol), **7** (1.14 g, 5 mmol), and a solution of sodium acetate (0.50 g, 6.0 mmol) in glacial acetic acid (20 mL) was refluxed for 140 °C for 18 h. After cooling, it was poured onto H_2_O (100 mL), and the solid formed was collected by filtration and washed with H_2_O and petroleum ether. The purple solid was further dried in vacuo, resulting in **9a** in 91% yield. 1H-NMR (500 MHz, DMSO-D_6_) δ 11.17 (s, 1H), 7.95 (td, J = 7.9, 4.0 Hz, 1H), 7.79 (d, J = 7.4 Hz, 1H), 7.74 (t, J = 8.9 Hz, 1H), 5.16 (dd, J = 12.9, 5.4 Hz, 1H), 2.85–2.91 (m, 1H), 2.58–2.63 (m, 1H), 2.52 (t, J = 2.0 Hz, 1H), and 2.04–2.07 (m, 1H).

#### 3.2.3. Synthesis of **9c**

A mixture of 36 mL glacial acetic acid and (0.308 g, 3.75 mmol) anhydrous sodium acetate was added to (0.58 g, 3 mmol) of 3-nitrophthalic anhydride (0.57 g, 2.5 mmol) of compound **7**. The reaction mixture was heated at 140 °C for 18 h. After completion of the reaction, the reaction mixture was cooled to rt, water was added to the reaction mixture, and it was stirred for another 30 min. The solid was filtered and washed with water. The product was dried under vacuum to obtain **9b** in 44.5% yield. ^1^H-NMR (500 MHz, DMSO-D6) δ 11.18 (s, 1H), 8.35 (d, J = 7.4 Hz, 1H), 8.24 (d, J = 6.9 Hz, 1H), 8.12 (t, J = 7.7 Hz, 1H), 5.20 (dd, J = 12.9, 5.4 Hz, 1H), 2.85–2.92 (m, 1H), 2.54–2.63 (m, 1H), 2.52–2.54 (m, 1H), and 2.05–2.10 (m, 1H).

To the mixture of compound **9b** (0.61 g, 2.02 mmol) in acetone was added a catalytic amount of 10% Pd/C (0.24 g, 0.66 mmol). The mixture was stirred at room temperature for 72 h under H2 gas. Then, the mixture was filtered to remove the palladium and acetone was evaporated to afford **9c** in 90.8% yield. ^1^H-NMR (500 MHz, DMSO-D_6_) δ 11.10 (s, 1H), 7.47 (dd, J = 8.3, 7.2 Hz, 1H), 7.01 (t, J = 7.4 Hz, 2H), 6.53 (s, 2H), 5.05 (dd, J = 12.9, 5.4 Hz, 1H), 2.84–2.89 (m, 1H), 2.52–2.60 (m, 2H), and 2.02 (td, J = 6.4, 2.1 Hz, 1H).

#### 3.2.4. Synthesis of Linkers (**12b** and **13b**)

Synthesis of **12b**. A solution of 4,7,10-trioxa-1,13-tridecanediamine (1.0 g, 4.5 mmol) in anhydrous DCM (15 mL) was treated with Boc_2_O (0.164 g, 0.75 mmol) and TEA (0.627 mL, 4.5 mmol) in DCM (22.5 mL). The reaction was stirred at room temperature for 4 h. The resulting yellow oil was purified by column chromatography, resulting in **10b** in 51.5% yield. ^1^H-NMR (500 MHz, CDCl3) δ 5.26 (s, 1H), 3.51–3.53 (m, 4H), 3.46–3.49 (m, 4H), 3.41–3.45 (m, 4H), 3.09 (q, J = 6.1 Hz, 2H), 2.69 (t, J = 6.6 Hz, 2H), 2.47 (s, 2H), 1.63 (td, J = 13.0, 6.7 Hz, 4H), and 1.31 (s, 9H).

A solution of **10b** (0.2 g, 0.7 mmol) and **9a** (0.14 g, 0.8 mmol) in DMF (2 mL) was treated with DIPEA (0.18 g, 1.4 mmol) under 90 °C for 12 h. The reaction mixture was poured into water and extracted with EtOAc. The combined organic phases were dried and concentrated under the reduced pressure. The crude product was purified to give compound **12a** in 40% yield. ^1^H-NMR (500 MHz, CDCl_3_) δ 8.41 (s, 1H), 7.48 (dd, J = 8.6, 7.4 Hz, 1H), 7.07 (d, J = 6.9 Hz, 1H), 6.92 (d, J = 8.6 Hz, 1H), 6.43 (s, 1H), 4.99 (s, 1H), 4.90 (q, J = 5.7 Hz, 1H), 3.57–3.69 (m, 11H), 3.52 (t, J = 6.0 Hz, 2H), 3.39 (q, J = 6.1 Hz, 2H), 3.21 (q, J = 5.9 Hz, 2H), 2.69–2.89 (m, 4H), 2.09–2.14 (m, 1H), 1.89–1.94 (m, 2H), 1.71–1.76 (m, 2H), and 1.42 (s, 9H).

Compound **12a** (0.154 g, 0.267 mmol) was dissolved in TFA (0.78 mL, 10.13 mmol) and stirred for 30 min. The excess of the acid was removed in vacuo, and the resulting product was dried under vacuum to give **12b** in quantitative yield. ^1^H-NMR (500 MHz, CDCl3) δ 9.45 (s, 1H), 7.60 (t, J = 8.0 Hz, 1H), 7.28 (d, J = 7.4 Hz, 1H), 7.15 (d, J = 8.0 Hz, 1H), 7.01 (s, 2H), 5.29 (s, 1H), 5.01 (dd, J = 12.6, 5.2 Hz, 1H), 3.63–3.72 (m, 13H), 3.45 (t, J = 6.0 Hz, 2H), 3.24 (q, J = 5.2 Hz, 2H), 3.10 (s, 1H), 2.98 (s, 1H), 2.89 (s, 1H), 2.72–2.83 (m, 2H), 2.17–2.19 (m, 1H), and 1.94 (dt, J = 26.7, 5.3 Hz, 4H).

Synthesis of **13b**. A solution of **10b** (0.2 g, 0.7 mmol) dissolved in THF (2 mL) was treated with DIPEA (0.18 g, 1.4 mmol) and stirred for 6 h, resulting in **11** in 44.5% crude yield, which was used without further purification. ^1^H-NMR (500 MHz, CDCl_3_) δ 5.22–4.71 (1H), 4.02 (d, J = 1.7 Hz, 2H), 3.57–3.67 (m, 10H), 3.53 (td, J = 6.0, 1.7 CDCl3 Hz, 2H), 3.43 (q, J = 5.9 Hz, 2H), 3.21 (d, J = 4.0 Hz, 2H), 1.86 (s, 1H), 1.83 (q, J = 5.9 Hz, 3H), 1.75 (t, J = 6.3 Hz, 2H), and 1.43 (d, J = 2.3 Hz, 9H).

The compound **11** (0.14 g, 0.8 mmol) and (0.108 g, 0.96 mmol) chloroacetyl chloride in Acetonitrile (2 mL) was treated with cesium carbonate (0.18 g, 1.4 mmol) at rt for 3 h. The reaction mixture was filtered, and the solvent was removed under reduced pressure. The crude product was purified by column to give compound 13a in 33% yield. ^1^H-NMR (500 MHz, CDCl_3_) δ 7.38 (dd, J = 8.6, 7.4 Hz, 1H), 7.09 (d, J = 6.9 Hz, 1H), 6.87 (d, J = 8.0 Hz, 1H), 6.73 (s, 1H), 5.03 (q, J = 5.9 Hz, 1H), 4.46 (dd, J = 22.6, 15.8 Hz, 2H), 3.60–3.62 (m, 5H), 3.55–3.57 (m, 5H), 3.49–3.53 (m, 5H), 3.35 (dd, J = 10.0, 6.0 Hz, 2H), 3.17 (t, J = 6.6 Hz, 2H), 2.92–2.97 (m, 1H), 2.83–2.86 (m, 1H), 2.69 (dd, J = 12.3, 4.3 Hz, 1H), 2.15 (t, J = 1.7 Hz, 1H), 1.71–1.78 (m, 4H), and 1.41 (d, J = 5.2 Hz, 10H).

Compound **13a** (0.2 g, 0.31 mmol) was dissolved in TFA (0.78 mL, 10.13 mmol) and stirred for 30 min. The excess of the acid was removed in vacuo, and the resulting product was dried under vacuum to give **13b** in quantitative yield. ^1^H-NMR (500 MHz, CD_3_OD) δ 7.45 (dd, J = 8.3, 7.2 Hz, 1H), 7.04 (d, J = 6.9 Hz, 1H), 6.99 (d, J = 8.6 Hz, 1H), 5.18 (dd, J = 12.9, 5.4 Hz, 1H), 4.44 (d, J = 7.4 Hz, 2H), 3.60–3.66 (m, 10H), 3.56–3.58 (m, 2H), 3.49 (t, J = 6.0 Hz, 2H), 3.28–3.29 (m, 2H), 3.09 (t, J = 6.3 Hz, 3H), 2.86–2.99 (m, 4H), 1.91 (dd, J = 6.3, 5.2 Hz, 2H), and 1.76 (t, J = 6.3 Hz, 2H).

#### 3.2.5. Synthesis of PROTACS (**14**–**16**)

Synthesis of **14**. The mixture of compound **4** (0.02 g, 0.07 mmol) and **9a** (0.024 g, 0.07 mmol) dissolved in 1.5 mL DMF was added with DIPEA (0.03 mL, 0.14 mmol). The reaction mixture was refluxed at 90 °C for 48 h. The DMF was removed by evaporation, and the crude was purified by MPLC to obtain compound 13 in 58% yield. ^1^H-NMR (500 MHz, ACETONE-D_6_) δ 12.24 (s, 1H), 10.96 (s, 1H), 9.98 (s, 1H), 8.26 (s, 1H), 8.11 (d, J = 9.2 Hz, 1H), 8.02 (s, 1H), 7.58 (dd, J = 11.7, 8.9 Hz, 2H), 7.48 (q, J = 7.4 Hz, 2H), 6.96 (dd, J = 22.6, 7.7 Hz, 2H), 6.36 (s, 1H), 6.10 (d, J = 9.2 Hz, 1H), 5.06 (q, J = 6.1 Hz, 1H), 3.53 (d, J = 5.7 Hz, 2H), 3.30 (d, J = 6.3 Hz, 2H), 2.91–2.98 (m, 4H), 2.71–2.79 (m, 2H), 2.16–2.21 (m, 1H), 1.73 (s, 4H). ^13^C-NMR (126 MHz, ACETONE-D_6_) δ 172.74, 170.36, 170.26, 168.22, 161.61, 153.33, 147.61, 138.72, 136.86, 135.95, 134.63, 133.51, 132.57, 124.43, 124.29, 119.40, 117.45, 114.27, 111.27, 110.75, 103.27, 49.80, 42.64, 41.92, 31.96, 27.30 23.38 ESI MS (*m*/*z*); 598.2148 [M + H]^+^.

Synthesis of **15**. The mixture of compound **3** (0.039 g, 0.12 mmol) and 12b (0.09 g, 0.18 mmol) dissolved in 20 mL CH_3_CN was added with TEA (0.05 mL, 0.36 mmol). The reaction mixture was stirred at RT for 24 h. The CH_3_CN was removed by evaporation, and the crude was purified by MPLC to obtain compound **15** in 47% yield. ^1^H-NMR (500 MHz, ACETONE-D_6_) δ 12.23 (s, 1H), 10.99 (s, 1H), 9.97 (s, 1H), 8.30 (s, 1H), 8.06–8.10 (m, 2H), 7.49–7.60 (m, 3H), 7.37 (s, 1H), 6.99 (dd, J = 26.1, 7.7 Hz, 2H), 6.56 (t, J = 5.4 Hz, 1H), 6.10 (d, J = 9.2 Hz, 1H), 5.05 (q, J = 6.1 Hz, 1H), 3.37–3.63 (m, 16H), 2.90–2.95 (m, 3H), 2.72–2.79 (m, 2H), 2.16–2.20 (m, 1H), 1.83–1.88 (m, 4H). ^13^C-NMR (126 MHz, ACETONE-D_6_) δ 172.75, 170.37, 170.14, 168.25, 161.54, 153.21, 147.73, 136.85, 135.85, 134.69, 124.22, 117.45, 113.99, 111.12, 103.39, 71.13, 71.09, 71.05, 70.86, 69.46, 49.78, 40.88, 23.39 ESI MS (*m*/*z*); 730.29 [M + H]^+^.

Synthesis of **16**. The mixture of compound **3** (0.056 g, 0.19 mmol) and **13b** (0.118 g, 0.22 mmol) dissolved in 20 mL CH_3_CN was added with TEA (0.03 mL, 0.23 mmol). The reaction mixture was stirred at RT for 18 h. The CH_3_CN was removed by evaporation, and the crude was purified by MPLC to obtain compound 16 in 38% yield. ^1^H-NMR (500 MHz, ACETONE-D_6_) δ 12.33 (s, 0H), 11.00 (s, 1H), 8.28 (s, 1H), 8.11 (d, J = 9.2 Hz, 1H), 7.45–7.63 (m, 4H), 7.26 (d, J = 14.9 Hz, 1H), 7.05 (dd, J = 13.7, 8.0 Hz, 2H), 6.20 (d, J = 7.4 Hz, 2H), 6.12 (d, J = 9.2 Hz, 1H), 5.15 (dd, J = 12.9, 5.4 Hz, 1H), 4.40 (s, 2H), 3.42–3.61 (m, 14H), 3.25 (tt, J = 20.4, 6.7 Hz, 2H), 3.05 (ddd, J = 18.3, 13.2, 4.6 Hz, 1H), 2.77–2.90 (m, 2H), 2.19–2.24 (m, 1H), 1.66–1.72 (m, 2H). ^13^C-NMR (126 MHz, ACETONE-D_6_) δ 171.92, 170.27, 169.85, 168.26, 167.23, 161.63, 153.31, 147.75, 136.28, 135.83, 134.61, 133.35, 132.60, 124.29, 122.34, 122.28, 119.31, 114.08, 112.25, 110.83, 110.59, 103.49, 71.06, 71.03, 70.82, 70.70, 69.46, 69.20, 50.32, 41.26, 37.38, 22.54. ESI MS (*m*/*z*); 787.31 [M + H]^+^.

### 3.3. Docking Studies

In silico docking of KRIBB11 with the 3D coordinates of the X-ray crystal structure of HSF1 (PDB code: 5d5U) was accomplished using the AutoDock 4.2 program downloaded from the Molecular Graphics Laboratory of the Scripps Research Institute. The AutoDock [22] program was chosen because it uses a genetic algorithm to generate the poses of the ligand inside a known or predicted binding site utilizing the Lamarckian version of the genetic algorithm, where the changes in conformations adopted by molecules after in situ optimization are used as subsequent poses for the offspring. In the docking experiments carried out, Gasteiger charges were placed on the X-ray structure of HSF1 along with KRIBB11, using tools from the AutoDock suite [23]. A grid box centered on the substrate binding pocket of HDACs enzyme with definitions of 50 × 50 × 50 points and 0.375 Å spacing was chosen for ligand docking experiments. The docking parameters consisted of setting the population size to 150, the number of generations to 27,000, and the number of evaluations to 2,500,000;the number of docking runs was set to 100, with a cutoff of 1 Å for the root-mean-square tolerance for the grouping of each docking run. The docking pose of HSF1 with compound KRIBB11 was depicted in Figure 1, and rendering of the picture was generated using PyMol (DeLanoScientific, South San Francisco, CA, USA) [24].

### 3.4. Cell Culture and Materials

Triple-negative breast cancer cells MDA-MB-231 were grown in DMEM high glucose and supplemented with streptomycin (500 mg/mL), penicillin (100 units/mL), and 10% fetal bovine serum (FBS). Cells were grown to confluence at 37 °C in a humidified atmosphere with 5% CO2. Antibodies for HSF-1 (1:1000; rabbit anti-human mAb; cat. no. 12972) and β-actin (1:1000; rabbit anti-human mAb; cat. no. 4970) were purchased from Cell Signaling Technology (Beverly, MA, USA).

### 3.5. MTS Colorimetric Assay

MDA-MB-231 cells were seeded at 2 × 103 cells per well in a clear 96-well plate, the medium volume was brought to 100 µL, and the cells were allowed to attach overnight. The cells were then incubated with the indicated concentration of compound 14, 15, or 16 at 37 ℃ for 24 h. Cell viability was determined using the Promega Cell Titer 96 Aqueous One Solution cell proliferation assay. Absorbance at 490 nm was read on Tecan Infinite F200 Pro plate reader, and values were expressed as percent of absorbance from cells incubated in DMSO alone.

### 3.6. Western Blot

Cells were seeded in 100 mm culture dishes (1 × 106 cells/dish) and allowed to attach overnight. The compound was added at varying concentrations, and the cells were incubated for an additional 24 h. For comparison, cells were also incubated with DMSO (0.5%) for 24 h. The cells were harvested in ice-cold lysis buffer (23 mM Tris-HCl pH 7.6, 130 mM NaCl, 1% NP-40, 1% sodium deoxycholate, 0.1% SDS), and 30 µg of lysate per lane was separated by SDS-PAGE, which was transferred to a PVDF membrane (Bio-Rad). The membrane was blocked with 5% skim milk in TBST and then incubated with the corresponding antibody (HSF-1 or β-actin). After binding of an appropriate secondary antibody coupled to horseradish peroxidase, proteins were visualized by ECL chemiluminescence according to the instructions of the manufacturer (GE healthcare, Chicago, IL, USA).

## 4. Conclusions

Keeping in mind the importance of HSF1 as a master regulator of heat shock protein and as a potential target for the development of anticancer drugs, we designed and synthesized new HSF1-based proteolysis-targeting chimeric molecules HSF1-PROTACs, in which the pomalidomide is tethered with HSF1 binder KRIBB11 to unlock the untapped potential and expand the repertoire of target warhead that can be recruited by PROTACs for anticancer drug development. Further evaluation to find better PROTACs with newer targets is underway in our lab and will be published in a due course of time.

## Data Availability

Not applicable.

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
