# Peer review of "Rational Design and Synthesis of HSF1-PROTACs for Anticancer Drug Development"

_molecules, 2022, doi:10.3390/molecules27051655_

Round 1
Reviewer 1 Report
This manuscript describes the design and synthesis of a PROTAC for HSF1. Unfortunately, the authors have not obtained any meaningful results here. The reviewer hopes that the compounds and data reported in this paper will be published in a future article with significant compounds. The reviewer believes that publication at this time is premature and that this manuscript deserves rejection.
Author Response
We thank the reviewer for reading our manuscript and for the critical feedback. We have designed our study to include the all-possible diversity in the linker for example like a hydrocarbon, polyethylene glycol type and one containing an amide functionality. Thus, the new HSF1-PROTACs were expected to give some insights to continue the biological studies in a meaningful direction. We have performed several sets of biological assays and a true discussion of the results have been presented. The details about cell cultures conditions and MTS colorimetric assay protocol are added in ‘Materials and methods’ for reproducibility purposes.
We believe our current design and multistep synthesis methodology could be exploited to make any number of new PROTACs. It is important to publish our findings in the current form so that anybody working on similar technique could do so with our protocol but with new HSF1 and E3 ligase ligands for making better PROTACs.
Reviewer 2 Report
In this manuscript, with the aim to provide new chemical agents to counteract the HSF1 overexpression-related resistance to cancer therapies, the authors made use of computational methods to design PROTAC molecules targeting HSF1. Three PROTAC molecules have been synthesized, exploiting a HSF1 inhibitor with different linkers. Effects on protein degradation and cell viability have been tested with Western Blot and with MTS colorimetric assay. Even if the results are not so relevant in a way to push towards a further validation of the newly synthesized molecules, the biochemical and cellular assays are poor. A deeper insight into the biochemical properties is expected prior to the studies of the effects on cells, and become necessary to address for a refinement of the molecules here presented. In a general view, the article can be useful as a starting point for the design, synthesis and flaws identification, of new PROTAC molecules targeting HSF1, but there are some overall weaknesses that the authors should consider prior to publication as reported next:
1. Line 57-58: is actually the inhibitor KRIBB11 used for tumor treatment? Perhaps a reference on that or a rewording of the phrase can clarify the point.
2. Line 59: A deeper description of the interactions that take place between KRIBB11 and HSF1 could be useful to: (i) better understand KRIBB11 binding mode; (ii) to more extensively create a link to Figure 1A, which shows some other interactions that are not reported in the text; (iii) to unambiguously clarify the rationale behind the design of the PROTAC molecules.
3. Line 101: Are there some additional data for molecules 14-16? (e.g. binding affinities or computational modeling)
4. Line 112: The data reported in the ‘Discussion’ section seem instead to be suited for the ‘Results’ one.
5. Line 126: perhaps further details about cell cultures conditions and MTS colorimetric assay protocol are needed to be added in ‘Materials and methods’ for reproducibility purposes.
6. Missing references: AutoDock4.2 (line 321), AutoDock suite (line 328), PyMOL (line 335)
7. In the manuscript there are a lot of syntactic and grammatical errors.
Author Response
In this manuscript, with the aim to provide new chemical agents to counteract the HSF1 overexpression-related resistance to cancer therapies, the authors made use of computational methods to design PROTAC molecules targeting HSF1. Three PROTAC molecules have been synthesized, exploiting a HSF1 inhibitor with different linkers. Effects on protein degradation and cell viability have been tested with Western Blot and with MTS colorimetric assay. Even if the results are not so relevant in a way to push towards a further validation of the newly synthesized molecules, the biochemical and cellular assays are poor. A deeper insight into the biochemical properties is expected prior to the studies of the effects on cells, and become necessary to address for a refinement of the molecules here presented. In a general view, the article can be useful as a starting point for the design, synthesis and flaws identification, of new PROTAC molecules targeting HSF1, but there are some overall weaknesses that the authors should consider prior to publication as reported next:
We thank the reviewer for the time spent in reading our manuscript and commenting on the manuscript. It has contributed to improving our manuscript. We have tried to address all of them in the best possible way.
- Line 57-58: is actually the inhibitor KRIBB11 used for tumor treatment? Perhaps a reference on that or a rewording of the phrase can clarify the point.
Yes, KRIBB11 was used for study involving tumors and reference 17 has been provided in line 59. - Line 59: A deeper description of the interactions that take place between KRIBB11 and HSF1 could be useful to: (i) better understand KRIBB11 binding mode; (ii) to more extensively create a link to Figure 1A, which shows some other interactions that are not reported in the text; (iii) to unambiguously clarify the rationale behind the design of the PROTAC molecules.
We have added from line 59 to 64: “The docking between the crystal structure HSF1 [18] and KRIBB11 revealed that the hydrogen bonding by the oxygen atoms of the nitro group of inhibitor stabilizes the binding to HSF1 protein. The nitrogen atoms of the pyridine ring and the N-methyl group of inhibitor form the hydrogen bonding interactions with the amide group of Gln72 and the phenol group of Tyr76, respectively. The N-methyl moiety in KRIBB11 is projected away from the protein as shown in figure 1.” - Line 101: Are there some additional data for molecules 14-16? (e.g. binding affinities or computational modeling)
We have provided all the publishable data in the manuscript and SI. We have no additional data.
- Line 112: The data reported in the ‘Discussion’ section seem instead to be suited for the ‘Results’ one.
We have renamed section 2, as Results and discussion. Line 54.
- Line 126: perhaps further details about cell cultures conditions and MTS colorimetric assay protocol are needed to be added in ‘Materials and methods’ for reproducibility purposes.
We have added related to cell culture, MTS assay and western blots in the manuscript from line 338 to 364.
- Missing references: AutoDock4.2 (line 321), AutoDock suite (line 328), PyMOL (line 335)
We have cited the text in section 3.4 with ref 22 -24 and added the references from line 429 to 435. - In the manuscript there are a lot of syntactic and grammatical errors.
We have checked the manuscript and have thoroughly revised the manuscript.
Reviewer 3 Report
Reviewer comments
This manuscript describes “Rational Design and Synthesis of HSF1-PROTACs for Anticancer Drug Development”. This is an interesting and well written article on design and synthesis of HSF1 inhibitor with pomalidomide. However, these compounds didn’t turn out to be promising but this study carried out nicely. I have two views on this manuscript (1) It would be better if authors can synthesize and test more compounds and add results in the manuscript; or (2) can perform some more biological study to draw better conclusion why these compounds are not good candidates for this target.
Other Major and Minor concerns:
- Performed assays to test its biological activity seems to be very limited. Therefore, more assays desired to test these molecules to draw conclusion based in this design.
- If authors like can add molecular docking study in this and try to corelate these results.
- It would be better if authors can acquire HRMS of lead compound 7i. As characterization is not very well established.
- %Purity and method purity analysis of tested compounds should be reported in manuscript.
Author Response
This manuscript describes “Rational Design and Synthesis of HSF1-PROTACs for Anticancer Drug Development”. This is an interesting and well written article on design and synthesis of HSF1 inhibitor with pomalidomide. However, these compounds didn’t turn out to be promising but this study carried out nicely. I have two views on this manuscript (1) It would be better if authors can synthesize and test more compounds and add results in the manuscript; or (2) can perform some more biological study to draw better conclusion why these compounds are not good candidates for this target.
We thank the reviewer for reading our manuscript and for very constructive feedback, which is much appreciated. We have designed our study to include the all-possible diversity in the linker for example like a hydrocarbon, polyethylene glycol type and one containing an amide functionality. Thus, the new HSF1-PROTACs were expected to give some insights to continue the biological studies in a meaningful direction. We have performed several sets of biological assays and a true discussion of the results have been presented. The details about cell cultures conditions and MTS colorimetric assay protocol are added in ‘Materials and methods’ for reproducibility purposes.
We would like to report our current design and multistep synthesis methodology which could be exploited to make any number of new PROTACs. It becomes more important to publish our findings in the current form so that anybody working on similar technique could do so with our protocol but with new HSF1 and E3 ligase ligands for making better PROTACs.
Other Major and Minor concerns:
- Performed assays to test its biological activity seems to be very limited. Therefore, more assays desired to test these molecules to draw conclusion based in this design.
We have performed several sets of biological assays and a true discussion of the results have been presented. Further, details about cell cultures conditions and MTS colorimetric assay protocol are added in ‘Materials and methods’ for reproducibility purposes. We have added related to cell culture, MTS assay and western blots in the manuscript from line 338 to 364.
- If authors like can add molecular docking study in this and try to corelate these results.
Molecular docking was used for design rationale and given in line 59 to 64. The docking between the crystal structure HSF1 [18] and KRIBB11 revealed that the hydrogen bonding by the oxygen atoms of the nitro group of inhibitor stabilizes the binding to HSF1 protein. The nitrogen atoms of the pyridine ring and the N-methyl group of inhibitor form the hydrogen bonding interactions with the amide group of Gln72 and the phenol group of Tyr76, respectively. The N-methyl moiety in KRIBB11 is projected away from the protein as shown in figure 1.” - It would be better if authors can acquire HRMS of lead compound 7i. As characterization is not very well established.
We have done ESI-MS of all lead compounds 14-16 and provided in line 294, 304 and 317 respectively.
%Purity and method purity analysis of tested compounds should be reported in manuscript.
%Purity was determined by HPLC in the range 98 to 100 % and the data has been added to the supplementary file Fig. S20.
Round 2
Reviewer 1 Report
With all due respect to the efforts of the authors, the compounds reported in this paper do not function as PROTACs, nor do they function as HSF1 inhibitors. In other words, the authors have not succeeded in rationally designing an HSF1-PROTAC. The reviewers believe that this compound and these results are unlikely to be published.
Reviewer 3 Report
Reviewer comments-revision
This manuscript “Rational Design and Synthesis of HSF1-PROTACs for Anticancer Drug Development”, after revision has improved somewhat. Authors has added HRMS and purity analysis data in manuscript. Also tried to explain results using molecular docking. Based on these improvements, I would recommend this manuscript for publication.